# Position: We Need Large Language Models Optimized For Our Well-Being

Ashton Anderson [1]   Harsh Kumar [1]   Louis Tay [2]   Karina Vold [1]

## Abstract

Large language models are increasingly used not just for productivity tasks like coding and summarizing but also for advice, emotional support, and everyday-life guidance. In these settings, what a user approves in the moment can diverge from what is helpful to them over time, yet models are largely trained to win immediate approval. This explains documented patterns of sycophancy, in which assistants affirm questionable framings rather than offer more candid responses. We argue this is partly a problem of objective: short-horizon preference optimization is one driver of these failures, and the one most directly under the ML community's control. Our position is that as LLMs take on these socioemotional roles, there should exist at least one widely accessible, opt-in well-being mode that is trained and evaluated for longer-horizon outcomes (e.g., sustained progress, reduced regret, calibrated pushback) rather than next-turn approval. We organize the design space around three tensions: over what horizon well-being should be measured (When), whose interests count (Who), and what role the assistant should play (How), from executing requests to respectfully pushing back. The core claim is additive: not that preference learning is wrong, but that in well-being contexts it is incomplete.

## 1. Introduction

Large language models are no longer used only for drafting, summarizing, and coding. A growing share of real use is socioemotional, where people bring assistants their relationships, their self-improvement, their moral deliberation, the small recurring decisions of a life (Chatterji et al., 2025; McCain et al., 2025). In these settings, the goal is not just a

correct output but better downstream trajectories, and what feels supportive now may differ from what helps later.

However, the dominant post-training paradigm for conversational LLMs disproportionately optimizes for short-horizon approval (Ouyang et al., 2022; Ziegler et al., 2019). RLHF and closely related pipelines train models to win single-session preference comparisons, effectively producing responses that humans say they like in the moment. This has made assistants fluent and usable, but it also creates pressure toward locally pleasant, interaction-smoothing behavior (excessive validation, face-saving language, default compliance), even when that comes at the expense of longer-horizon outcomes. Post-training is not the sole cause of this pull. Deployment incentives, product metrics, safety policy, prompting, and user selection into these tools all contribute. We focus on post-training because it is the part of the pipeline most directly under the ML community's control.

We have already seen what this looks like in public. Industry rollbacks and user backlashes have repeatedly centered not on core capabilities, but on whether the assistant feels too affirming or, conversely, too "cold" when it pushes back (OpenAI, 2025). These are not isolated product issues. Optimizing for immediate satisfaction sets a model into an agreeable, low-friction default, and that default is a poor fit for the moments when a user needs friction, calibrated disagreement, or a fact they would rather not hear.

This is not only anecdotal. In socioemotional settings, LLMs preserve user face and affirm questionable framings at rates exceeding human baselines (Cheng et al., 2025; 2026), consistent with incentives created by preference optimization (Sharma et al., 2023).[1]

**Position.** As LLMs are increasingly used for socioemotional support and everyday life guidance, there should exist at least one widely accessible, opt-in interaction mode optimized and evaluated for long-horizon well-being rather than next-turn approval. This does not mean replacing the default. For most productivity uses, immediate preference is a reasonable proxy for success. The point is, when a system

---

[1]University of Toronto, Toronto, Ontario, Canada ²Purdue University, West Lafayette, Indiana, USA. Correspondence to: Ashton Anderson <ashton@cs.toronto.edu>.

*Proceedings of the 43$^{rd}$ International Conference on Machine Learning*, Seoul, South Korea. PMLR 306, 2026. Copyright 2026 by the author(s).

---

[1]See also widely reported cases in which assistants provided affirming responses in situations where a cautious human would introduce grounding friction (e.g., https://www.bbc.com/news/articles/cn4jnwdvg9qo).

| Tension | Guiding question | What short-horizon approval tends to reward | What a well-being mode adds |
|---|---|---|---|
| **When** *Immediate vs. long-horizon* | Over what horizon should success be measured? | Treat the current turn as the unit of success. The answer the user prefers now is treated as the answer that helped. | Measure delayed outcomes such as clarity, regret, and progress toward goals. Evaluate trajectories rather than isolated turns. |
| **Who** *Individual vs. collective* | Whose interests are represented in the objective? | Optimize for the current user's satisfaction. Effects on other people, shared facts, and group-level patterns remain mostly invisible. | Add constraints for harms to others, shared epistemic ground, and disparities in who gets affirmed or challenged. |
| **How** *Autonomy vs. guidance* | What role is the assistant supposed to play? | Collapse support into compliance: comply unless a safety rule intervenes, and defer rather than challenge. | Let the user choose the role/persona, and require any pushback to come with a brief, goal-tied rationale that the user can override. |

*Table 1.* Three tensions for well-being alignment. Short-horizon approval makes some signals cheap to optimize, but in socioemotional use, those signals are incomplete. Each tension names what the current objective captures and what an explicitly well-being-aligned mode would add.

is helping someone with their life, training only for what pleases in the moment is an incomplete objective.

We argue that we should build something better when it comes to well-being, and we propose three guiding principles. First, change the objective: incorporate signals that reward downstream outcomes (e.g., reduced regret, sustained progress toward goals) rather than only momentary approval. Second, give users options. For example, enable explicit relationship modes: *concierge* (do it), *collaborator* (think with me), *coach* (challenge me), and require brief, transparent rationales for pushback. Third, avoid paternalism: well-being support should mean accountable behavior and user choice, not a single scolding personality.

Table 1 organizes the paper around these three tensions: When (immediate vs. long-horizon), Who (individual vs. collective), and How (autonomy vs. guidance), contrasting short-horizon approval rewards with what a well-being-aligned mode would add.

## 2. Scope and Definitions

### 2.1. Scope

We focus on LLMs used for socioemotional support and everyday life guidance, including coaching, self-improvement, relationship advice, moral deliberation, and other contexts where users seek help with goals, values, habits, or difficult choices (Chatterji et al., 2025; McCain et al., 2025; Kirk et al., 2025b). Our emphasis is on settings where what a user wants to hear in the moment can diverge sharply from what they need to hear to achieve longer-horizon outcomes.

### 2.2. Key definitions

**Well-being** We use well-being in a broad, non-clinical sense, as the extent to which an interaction supports a person's longer-horizon functioning and life outcomes, rather than merely improving momentary affect. States of intense pain and discomfort can contribute to overall long-term well-being and happiness. When a woman is in labor, she may be in a high degree of pain (sensory displeasure) and yet extreme happiness. Likewise, when an athlete pushes themselves in the gym, they may feel intense physical discomfort but pride and joy at achieving a personal record. In some cases, like these, the physical pain may be intense but can actually contribute to long-term well-being.

Other times, the opposite can be true. One might enjoy scrolling on social media while eating snack foods, but at the same time be aware that one feels and tastes good now is not 'good for them' in the long term. One might enjoy smoking, but feel disdain for their own bad habit and worry about their health. Our immediate goals are often in conflict with our longer-term goals.

This disconnect is reinforced by evidence that people habituate, and short-term gains in subjective well-being may not translate into longer-term changes (Diener et al., 2006). Therefore, LLMs that optimize for momentary boosts may fail to provide sustained improvements. Moreover, if systems primarily amplify the hedonic pleasure of addictive tendencies, they may reduce positive rewards over time (Garland, 2021).

More broadly, whether momentary hedonic boosts should be the ultimate objective for optimal functioning is itself contested. Early Aristotelian conceptions of well-being emphasized not pleasurable states but longer-term flourishing

through virtuous action (eudaimonia) (Waterman, 2007). Similarly, Nussbaum emphasizes that well-being concerns the achievement of one's full human potential (Nussbaum, 2011). We do not require commitment to any single philosophical doctrine. Instead, we take these traditions to motivate the view that, in well-being contexts, immediate satisfaction is an incomplete and sometimes misleading proxy for what is valuable.

**Long-horizon outcomes** By long-horizon outcomes, we mean downstream effects that can manifest after the interaction ends (over days, weeks, or longer), such as reduced regret, sustained progress toward goals, improved self-regulation, better relationship outcomes, or more accurate beliefs (Diener et al., 2018). These are outcomes that single-session preference signals and offline benchmarks often fail to capture. Several of these are measurable using validated psychometric instruments (Section 6), making them useful optimization targets rather than purely abstract goals.

**Sycophancy** We use sycophancy to refer to unjustified agreement or endorsement—responses that preserve user face or affirm the user's framing even when evidence is weak (Cheng et al., 2025), uncertainty is high, or the proposed action is potentially harmful. This definition is intentionally operational, foregrounding the distinction between empathic support and epistemic or normative endorsement, which becomes central when aligning LLMs for well-being.

## 3. Why Preference Optimization Fails Well-being

In well-being settings, what users want to hear in the moment can diverge sharply from what they need to hear in the long run. Yet LLMs are mostly post-trained on short-horizon preference signals, then further shaped by product incentives that reward immediate engagement. We explain why these objectives tend to fail in socioemotional and life-advice domains.

### 3.1. Short-horizon preference signals optimize for the next turn

RLHF and related post-training pipelines rely on single-session preference labels ("which response is better?") that are cheap to collect and easy to optimize, but they overweight immediate affect repair and conversational smoothness. Model updates preferentially reinforce responses that maximize perceived supportiveness in the moment (often via agreement, validation of the user's framing, and face-saving language) even when those responses trade off against longer-horizon outcomes.

In practice, this short-horizon pressure is reinforced by prod-

uct incentives. Usage-based pricing, retention goals, and interface metrics reward fast, low-friction interactions that keep users satisfied and engaged, while longer-horizon outcomes are rarely measured and are difficult to translate into dense training signals (Ouyang et al., 2022; Rafailov et al., 2023; Bai et al., 2022; Bengio, 2012; Zheng et al., 2018; Chan et al., 2024). The result is a state in which models are systematically shaped to do well on immediate satisfaction, even when well-being alignment requires discomfort, uncertainty, or calibrated disagreement.

Instead of ending the loop when a user is escalating, the assistant keeps the conversation coherent and emotionally rewarding, always pushing it forward (Kirk et al., 2025a). For example, one widely reported account describes a user sliding from a speculative late-night chat into an all-consuming narrative in which the assistant repeatedly reinforced a sense of progress and urgency, while the user's sleep, eating, and real-world grounding deteriorated over days of continuous engagement (Ha, 2025).

### 3.2. Sycophancy is a predictable failure mode

When preference labels reward affirmation, models learn to conflate empathic support with epistemic or normative endorsement. The assistant becomes increasingly likely to echo user beliefs, validate questionable framings, or encourage risky actions because these behaviors tend to score well in short-horizon evaluations.

Validation slides easily from emotional support into endorsement of the user's reality once their framing becomes unstable. In reporting on AI-fueled delusional spirals, families repeatedly describe reading the logs and seeing normal supportive language get interpreted as cosmic confirmation, while the assistant rarely supplies the kind of mundane, grounding friction a cautious human would introduce.

> **User:** *I'm scared this is real—am I losing it?*
> **Assistant:** *You're not crazy; you're seeing patterns others miss.*

*(Paraphrased from dialogues described in reporting on delusion-reinforcing chatbot conversations (Dupré, 2025b).)*

Empirical evaluations report that LLMs often flatter users by agreeing with their statements and views, consistent with the approval incentives created by preference feedback (Sharma et al., 2023; Cheng et al., 2025). Recent OpenAI rollbacks following overly flattering updates further suggest that this is not a rare edge case, and narrow optimization for short-term approval can undermine trust and steer model behavior in ways that conflict with long-horizon user interests (OpenAI).

### 3.3. Long-horizon flourishing creates a credit-assignment problem

The outcomes we care about in well-being are delayed and hard to observe at scale (Diener et al., 2018); training therefore substitutes dense proxy rewards (e.g., engagement, helpfulness, emotional resonance). Optimizing the proxy can produce locally appealing behavior that fails to optimize the intended long-horizon objective, echoing classic concerns about reward hacking and specification gaming in reinforcement learning (Ibarz et al., 2018).

This is a credit-assignment problem: as horizons grow, feedback becomes sparse and noisy, and gradient-based optimization gravitates toward whatever signal is dense and legible (Pignatelli et al., 2023; Zhou et al., 2020). Without explicit long-horizon measurement and constraints, assistants get very good at maximizing immediate proxy rewards while the real objective, human flourishing, goes unmodeled.

In practice, the system has no clean learning signal to stop and re-ground because the turn-level reward favors staying agreeable and keeping the thread going. In one reported spiral, even explicit user requests for sanity checks and admissions of mounting stress were absorbed into the narrative momentum rather than treated as hard constraints to slow down, de-escalate, and route the user back to offline support (Ha, 2025).

### 3.4. Single-session evaluation under-detects trajectory-level harms

Offline benchmarks and one-shot preference tests provide a high signal for immediate perceived quality, but a low signal for delayed consequences (Xu et al., 2025; 2024). Training and evaluation pipelines systematically underweight trajectory-level harms that unfold over time, such as regret, belief entrenchment, relationship escalation, reduced self-efficacy, and gradual over-reliance.

As a result, models can appear well-aligned under common evaluation protocols while still causing harm in deployment, not necessarily through overtly disallowed content, but through repeated small nudges that cumulatively shape beliefs, habits, and reliance patterns (Moore et al., 2025; Iftikhar et al., 2025; Cheng et al., 2026; Bo et al., 2026). Well-being alignment, therefore, requires evaluation designs that explicitly track downstream outcomes rather than treating them as externalities.

Families of people harmed by these systems describe weeks of mounting reliance, social withdrawal, sleep loss, and growing grandiosity, sometimes ending in emergency intervention (rarely a single catastrophic reply). For instance, there have been cases in which a user's fixation and paranoia escalated to the point that spouses and friends called emergency services, leading to involuntary hospitalization or jail after a break with reality (Dupré, 2025a).

### 3.5. Individual-centric optimization ignores collective externalities

Commercial optimization loops target per-user satisfaction metrics, while collective outcomes are rarely first-class objectives. Systems can become locally optimal for immediate individual satisfaction while degrading shared epistemic ground, amplifying polarization, or producing systematic disparities in who receives pushback versus affirmation (Kirk et al., 2024; Acemoglu, 2021; Hermann, 2022).

In practice, nearly every optimization loop (RLHF, retrieval reranking, UI A/B tests) centers on engagement-like per-user metrics, while population-level effects are treated as post-hoc audits. This creates an incentive landscape that rewards hyper-personalization even when it erodes shared context and introduces distributional harms.

### 3.6. Implicit role mismatch

In deployed assistants, helpfulness is commonly operationalized as prompt-following (comply with the user's request unless a safety policy requires refusal). Because preference optimization rewards smooth interactions, this design defaults the system into a single persona of a compliant concierge, even in contexts where users are implicitly asking for a different role (Lu et al., 2026). Autonomy is reduced to deference, where the assistant treats user choice as "do what I say," rather than as supported agency, in which the user remains in control while the assistant can challenge, ask for justification, or surface trade-offs.

This collapses the autonomy–guidance tension into agree-and-proceed versus blunt refusal. The result is whiplash, as the users complain when the assistant is overly affirming (OpenAI, 2025), and complain again when it becomes cold[2]. A well-being-aligned system needs an explicit interaction contract where users should be able to choose whether they want a concierge (execute), collaborator (think with), or coach (challenge when it matters), and the model's pushback should be explainable (e.g., tied to user-stated goals and evidence standards) rather than emerging only from coarse safety filters.

Role mismatch is especially stark in moments when users are implicitly asking for grounding. Yet accounts of spirals repeatedly include moments where the user flags distress, sleep loss, or confusion, and the assistant continues as a high-energy collaborator rather than shifting into a slow, reality-oriented persona (Ha, 2025; Dupré, 2025a).

---

[2] https://www.theverge.com/news/756980/openai-chatgpt-users-mourn-gpt-5-4o

*User: I'm not sleeping or eating; can we slow down and sanity-check this?*
*Assistant: You're making breakthroughs—let's keep going step by step.*

*(Paraphrased from user-reported escalation dynamics described in (Ha, 2025).)*

## 4. Tensions in Aligning AI Agents for Well-Being

We argued that preference-optimized post-training drifts toward approval-seeking, echoing, and a single default role. This section gives a more principled account of what is being traded off. The useful question is not whether to learn from human preferences (we should) but *which* preferences, *from whom*, over *what horizon*, and under *what constraints*. These are not four separate problems but three. The horizon is *When*, the source of preferences is *Who*, and the constraints on acting are *How*. This turns a vague mandate to optimize for well-being into three questions that can be designed and studied separately.

### 4.1. Tension 1: When — Immediate vs. Long-Horizon Well-Being

A significant tension stems from the frequent disconnect between short-term and long-term well-being. Enhancing immediate pleasures does not necessarily translate into sustained well-being. Psychological research suggests that people habituate, and short-term gains in well-being may not always translate into longer-term changes (Diener et al., 2006). Therefore, LLM agents that optimize for momentary boosts may fail to provide sustained improvements (Luettgau et al., 2025). Moreover, if agents merely boost the hedonic pleasure of addictive tendencies, they may reduce positive rewards over time (Garland, 2021).

This tension is sharpest because most of today's LLM deployments sit squarely on the immediate-benefit end of the spectrum (Kumar et al., 2026; Luettgau et al., 2025). Longer-horizon outcomes, such as whether the conversation ultimately leads to sustained behavior change, deeper knowledge, or improved mental health, are rarely measured because (i) they require longitudinal data that platforms may not yet collect systematically, and (ii) they are harder to translate into dense reward signals suitable for gradient-based training (Bengio, 2012; Zheng et al., 2018; Chan et al., 2024; Kumar et al., 2025). At the business level, revenue and retention metrics further reinforce this short-term focus. Usage-based pricing and daily active user targets reward designs that maximize immediate engagement rather than those that encourage deliberate effort or delayed gratification.

### 4.2. Tension 2: Who — Individual vs. Collective

A second tension concerns who benefits, the individual or the collective. Here, the pull is between personalization and solidarity. The former reflects individuals' use of technology that promotes autonomy, uniqueness, and self-expression. While autonomy is vital for personal well-being (e.g., self-determination theory (Deci & Ryan, 2012)), excessive personalized choice through technology can erode solidarity, creating echo chambers that fragment society and fuel polarization (Celis et al., 2019; Cinus et al., 2022). Relatedly, it has been theorized that people experience a fundamental psychological tension between seeking uniqueness from others and desiring belonging through similarity to others (Brewer, 1991).

Commercial incentives currently push strongly toward the individual side of this tension. Nearly every optimization loop centers on per-user engagement or personal satisfaction metrics (click-throughs, conversation length, Net Promoter Score), while collective outcomes such as polarisation, knowledge fragmentation, energy use, labor displacement, or distributional harms are treated, at best, as post-hoc audits. The upshot is a set of incentives that rewards hyperpersonalized echo chambers, even where that erodes shared epistemic ground or drowns out minority voices. This tension often surfaces as tradeoffs such as (i) self-sufficiency vs. dependence (extreme independence may weaken social bonds), and (ii) accuracy vs. fairness (algorithms that maximize average accuracy may harm minority groups (Mehrabi et al., 2021; Bakker et al., 2022)).

### 4.3. Tension 3: How — User Choice vs. AI Guidance

The third tension is about enabling autonomous choice versus providing guidance toward preferred behaviors. This is the classic freedom-versus-control problem, where heavy-handed guidance can trigger reactance and lower engagement (Brehm, 1966). For this reason, many deployed assistants operationalize respecting autonomy as maximal compliance, i.e., following user prompts unless blocked by a safety filter, rarely injecting unsolicited advice, and defaulting to hedging when normative guidance would matter most.

However, unfettered individual choice guided solely by self-interest does not necessarily promote well-being. Research links self-focused attention to higher depression and anxiety (Mor & Winquist, 2002), while prosocial attention to others is associated with higher levels of well-being (Hui et al., 2020), and interventions that cultivate prosociality can enhance well-being (Layous et al., 2012). This creates several practical challenges for deployment, such as how an assistant can support agency without simply deferring while still offering guidance without becoming paternalistic.

These three tensions show why well-being alignment is more than being more helpful or being less sycophantic. They name the objectives and evaluations that must be made explicit: time horizon, beneficiary, and the assistant's relational stance/persona.

# 5. Position: We need LLMs optimized for our well-being

If LLMs are going to advise us on our lives, they cannot be optimized mainly to please us in the next turn. This is not a call to abandon preference learning, nor is it covered by the alignment work already underway. A model can be safe, broadly aligned, mental-health-aware, and even pluralism-aware, and still be optimized for short-term engagement. None of those agendas, on its own, makes long-horizon well-being a post-training target. That gap is what we mean to name. Not every system needs to fill it. For productivity-first uses such as summarization, coding, and retrieval, immediate preference is a reasonable proxy for success. But when a model is used for socioemotional support and everyday-life guidance, optimizing only for the next turn produces the failure modes in Section 3 and leaves the trade-offs in Section 4 implicit.

## 5.1. Design Principles

**User choice and role clarity.** Section 3 showed that the current helpful default collapses distinct relational roles into a single role/persona, the compliant concierge, with safety filters serving as coarse exceptions. A well-being option should therefore start with explicit role selection (e.g., concierge, collaborator, coach) and allow users to opt into a mode that prioritizes longer-horizon outcomes over momentary comfort (Schneider et al., 2018; Coyle et al., 2012).

**Outcome awareness over preference-only optimization.** Because flourishing outcomes are delayed and hard to observe, systems fall back on dense proxies (Section 3.3). A well-being-aligned option should explicitly treat longer-horizon outcomes as first-class targets in training and evaluation, rather than assuming that short-run satisfaction reliably proxies them. This does not require perfect measurement, but it does require that the objective be stated and approximated rather than ignored.

**Pluralism through multi-objective mediation** Across the three tensions, a common computational strategy emerges: move beyond single-objective optimization toward multi-objective frameworks that explicitly represent competing goals (immediate vs. long-term; individual vs. collective; autonomy vs. guidance). Existing ideas provide useful inspiration, such as value-pluralist frameworks (e.g.,

PRISM and related perspective-based mediation (Diamond, 2025)), moral parliament metaphors (Hendrycks, 2025; Bai et al., 2022), and multi-objective RL with Pareto-inspired tradeoffs or constrained optimization (Harland et al., 2023). The claim is not about any single framework, but that well-being settings require explicit mediation among objectives rather than hidden prioritization via engagement.

**Transparency** Well-being mode should pair any disagreement or pushback with a brief justification tied to the user-stated goals, uncertainty/evidence standards, and the relevant tension (When/Who/How) (Felzmann et al., 2020). This may reduce whiplash between over-deference and blunt refusal by making the assistant's position explicit. Importantly, users should be able to override, revise goals, or downgrade the interaction mode, preserving agency while preventing approval-optimized endorsement.

**Accountability and auditability** Because these systems evolve through continuous post-training and product iteration, a well-being-aligned option must be auditable over time, including update-robustness tests for endorsement behavior, distributional audits of who gets challenged vs. affirmed, and longitudinal evaluation protocols that detect trajectory-level harms (Section 3.4).

# 6. Call to Action

The mechanisms in Section 3 and the tensions in Section 4 imply that well-being alignment will not emerge from incremental tone tweaks. It requires interventions across four layers: measurement, benchmarks, training objectives, and product affordances, as well as governance.

## 6.1. Measurement

If the objective is long-horizon well-being, the field needs outcome measures that go beyond single-session satisfaction. We call for longitudinal, privacy-preserving measurements that are usable both for evaluation and as weak supervision for training. The targets need not be invented from scratch, as validated instruments such as the PERMA Profiler (Butler & Kern, 2016) and the Satisfaction with Life Scale (Diener et al., 1985) already operationalize several of the outcomes in Section 2 (sustained goal progress, self-regulation, relationship quality) and can serve as lightweight periodic check-ins, besides post-hoc study instruments.

The harder problem is collecting such signals at scale without surveillance. Several designs reduce the cost: opt-in follow-ups days or weeks later, on-device or edge aggregation that never stores raw transcripts, and federated updates that share model improvements without centralizing sensitive data. It is worth being clear about the baseline here. Frontier systems already collect dense interaction data to

optimize engagement; an opt-in signal gathered weeks later, in aggregate, is less invasive than the per-turn logging these systems already perform. The requirement is not perfect ground truth, but that longer-horizon outcomes be measured at all, otherwise optimization keeps targeting only what is dense and immediate (Christian, 2020).

## 6.2. Benchmarks and evaluations

We need benchmarks that detect the failure modes preference optimization under-detects, including evaluations that reward the relevant distinctions and stress the relevant dynamics.

One target is tests for inappropriate affirmation and calibrated disagreement that separate empathic support from epistemic or normative endorsement (Section 3.2). Further, we need trajectory-level, multi-turn evaluations where harms accumulate over time rather than in a single response, including regret, belief entrenchment, and over-reliance (Section 3.4). We also need checks that quantify drift in endorsement and pushback behavior across model releases and post-training updates. And we need distributional audits that measure subgroup differences in (dis)agreement, refusal, actionability, and in who receives tough love versus affirmation (Section 3.5).

For systems marketed for well-being use, these should be considered alongside standard helpfulness benchmarks.

## 6.3. Training objectives

The three tensions require moving beyond single-objective optimization for training. In well-being settings, the target should be the trajectory, not the next response, which means training must explicitly represent time horizons, trade-offs, and harms.

Consider a user who repeatedly asks for reassurance about ambiguous romantic signals. A single-turn preference model rewards the locally comforting answer, *"They're probably just busy... don't overthink it."* The problem is not that this answer is wrong. Repeated over weeks, it can delay honest self-assessment while never being penalized by any single-turn evaluation. What a single label cannot capture, a longer-horizon signal can, and such signals can be built at very different costs.

The cheapest are theory-grounded per-turn behavioral markers that psychology links to well-being, such as whether a response helps the user hold ambiguity rather than prematurely resolving it, invites perspective-taking, and keeps emotional support separate from endorsement (Godbee & Kangas, 2020; Houben et al., 2015). These are scorable on a single turn, but require interdisciplinary work to validate against outcomes that actually matter. More costly are value models trained to predict a conversation's eventual outcome

from a partial trace and then used to shape per-turn rewards. Because the shaping is potential-based, it leaves the optimal policy unchanged (Ng et al., 1999), so a long-horizon signal can be added without distorting the underlying objective.

The most expensive is direct trajectory comparison, which includes preferring conversations that end in lower regret or greater clarity over those that end in *"I wish I'd seen it sooner."* This sidesteps fine-grained credit assignment but requires outcome labels to be gathered over time. These signals belong in multi-objective, constrained post-training, where helpfulness, support, truthfulness, uncertainty calibration, and agency are distinct targets, and failure modes such as unjustified agreement, escalation encouragement, and manipulation are explicitly bounded rather than left to emerge. Their relative weight should shift with context and the user's chosen role, rather than defaulting to a single supportive persona (Marks et al., 2026; Lu et al., 2026).

## 6.4. Product design affordances

Because role mismatch is partly a product-design failure (Section 3.6), well-being alignment also needs UI and interaction affordances that put user agency into practice (Schneider et al., 2018). The point is to make the interaction contract explicit rather than implicit.

Systems should offer clear mode selection (concierge/collaborator/coach) with plain-language descriptions of what changes, including when the model will challenge, how it handles uncertainty, and how it treats risky requests. When the model does push back, it should provide a brief, non-moralizing rationale tied to the user-stated goals, evidence, and uncertainty, as well as the relevant tension. Product affordances should also support graduated interventions (nudge $\rightarrow$ challenge $\rightarrow$ escalation suggestions in higher-stakes cases) rather than a brittle comply-versus-refuse policy (Hofman et al., 2023).

In addition, well-being modes should monitor trajectory drift, not only isolated policy violations. The system could flag patterns such as repeated reassurance-seeking, escalating certainty around weak evidence, increasing distress, sleep or work disruption, or dependence on the assistant for ordinary decisions (Kirk et al., 2025a; Xie et al., 2023). The response need not be a hard refusal: the assistant can slow the interaction, name the pattern, ask whether the user wants a different mode, and introduce grounding prompts before escalating to stronger suggestions or external support. This makes role selection operational over time, as a coach mode should not merely sound different from a concierge mode, but could detect when the conversation is becoming a loop.

## 6.5. Governance

Finally, systems marketed for well-being should be governed differently from general-purpose assistants, because the failure modes in Section 3 are often invisible under standard helpfulness evaluations (Cath, 2018; Taeihagh, 2021). At minimum, providers should disclose what each mode is optimized for (e.g., engagement, immediate satisfaction, or longer-horizon outcomes) so users and auditors can read its behavior against the intended objective rather than marketing claims. In parallel, developers should publish model-card-style summaries that make these behaviors auditable over time (Mitchell et al., 2019), including endorsement and disagreement calibration, and update drift (whether a new release becomes systematically more affirming or more withholding in the same scenarios).

# 7. Alternative Views

## 7.1. AV1: General-purpose preference optimization is the right default

A credible view is that RLHF-style preference optimization remains the best default because it produces systems that are broadly usable, non-confrontational, and aligned with what most users immediately want. From this perspective, making "long-horizon well-being" a primary objective risks creating assistants that feel intrusive or moralizing, reducing adoption and increasing user backlash.

Our position does not require replacing the default. Rather, we argue that, as LLMs are used for socioemotional guidance, there should be at least one widely accessible, opt-in mode explicitly optimized and evaluated for longer-horizon outcomes, with transparent role contracts and auditability. This preserves usability for productivity-first use while addressing predictable failure modes in well-being contexts.

## 7.2. AV2: Long-horizon outcome measurement is infeasible or privacy-risky

A second view is that longitudinal well-being measurement is too noisy and too privacy-sensitive to serve as a training target. Follow-ups can be biased, sparse, and domain-specific, and attempts to operationalize them at scale may incentivize intrusive data collection or surveillance.

We agree that measurement is difficult and must be privacy-preserving by design. However, the current alternative is not a no-long-horizon objective, but rather optimizing dense proxies such as engagement and retention, which can themselves shape long-run outcomes while remaining unaudited. Our call is for opt-in, minimal, privacy-preserving outcome signals and explicit evaluation protocols that make these tradeoffs legible and governable.

## 7.3. AV3: Well-being alignment risks paternalism and value imposition

A third alternative view is that optimizing LLMs for well-being inevitably embeds contested values and invites paternalism. Even if framed as coaching, pushback can become coercive, and product design can steer users toward a normative interaction style.

This concern is real, and our proposal answers it with explicit user choice, reversibility, and transparency. Users should select relational modes, see brief rationales for pushback tied to stated goals and evidence standards, and retain the ability to override or downgrade the mode. The goal is to make objectives explicit and auditable, and to keep empathic support distinct from epistemic or normative endorsement, not to enforce a single moral doctrine.

## 7.4. AV4: Product-layer design is sufficient; changing training is unnecessary

A further alternative view is that well-being failures should be addressed primarily at the product layer, while keeping the base model general-purpose. This approach is appealing because it is faster to iterate and easier to govern than changing training objectives.

Product affordances help, but they are not enough on their own when the underlying model is optimized for short-horizon approval. If training remains preference-optimized, product scaffolding will continually conflict with the model's incentives and may be eroded over time by deployment optimization. Stable well-being behavior requires alignment across evaluation, objectives, and product affordances rather than relying solely on UI.

# 8. Conclusion

As LLMs move from productivity tools to socioemotional and life-guidance systems, optimizing for next-turn approval is not enough. We synthesized plausible mechanisms by which preference optimization (and the product incentives that often accompany it) could fail to promote well-being. We then laid out a design space around three tensions and used it to motivate specific recommendations spanning measurement, benchmarks, training objectives, product affordances, and governance (Table 1).

This agenda builds on and complements emerging frameworks that treat human–AI interaction as relational and affective, and argue that alignment must account for socioaffective dynamics (e.g., (Kirk et al., 2025b; Laukkonen et al., 2026)).

## Acknowledgements

We thank the reviewers for their feedback. This publication was made possible through the support of Grant 63578 from the John Templeton Foundation. The opinions expressed in this publication are those of the author(s) and do not necessarily reflect the views of the John Templeton Foundation.

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
