# OpenReview forum: "Position: We Need Large Language Models Optimized For Our Well-Being"
_ICML.cc/2026/Position_Paper_Track — ICML 2026 Position Paper Track regular_

### Official Review · Reviewer_4eZ4 · 2026-03-08

**Significance:** 3
**Argument Clarity:** 4
**Rating:** 6
**Confidence:** 4

**Questions:**

1. How do we know that sycophancy will be resolved if we optimize for well-being? Meaning, how do we know that sycophancy is just a symptom of the short-run preferences of individuals?

**Alternative Views Section:**

Yes

**Compliance With Llm Reviewing Policy A Conservative:**

Affirmed.

**Discussion Potential:**

4

**Final Justification:**

I think the paper is quite strong, and would advocate for it to be accepted.

**Paper Summary:**

The authors claim that the current paradigm of LLM development does not account for longer term well-being. That is, patterns such as sycophancy arise because we fail to account for the longer-term implications of the decisions made by LLMs. Instead, we should design LLMs so they optimize for the long-term reward, thereby prioritizing things that might be less preferred in the short-run, but more preferred in the long-run. Current preference optimization-based approaches fail to accomplish this because they consider only a short-horizon, and also fail to distinguish between scenarios where the AI is a "coach" vs. those where the AI is trained to instruct. To accomplish this, the authors propose that we focus on longer-term evaluation of LLMs, along with long-term training objectives and affordances that allow people to control the degrees of intervention.

**Position:**

Yes

**Position In Title:**

Yes

**Related Work:**

2

**Strengths And Weaknesses:**

## Strengths
1. **Well-Written Argument** - The authors present a well-written argument in favor of optimizing LLMs for individual's well-being. The authors clearly define key terms such as well-being and long-horizon, and use this to build up a step-by-step argument of the shortcomings of the current paradigm for training LLMs, and how a future paradigm can fix these issues. Especially helpful is the author's organization of the paper, as it clearly proceeds by first describing current issues and tensions that need to be resolved, then proceeding to a broader call to research.
2. **Clear Next Steps and Call to Research** - Section 5 and 6 in the article are helpful in being a clear call to research. The authors clearly list out the set of research that needs to happen to accomplish their goal of optimizing for well-being, including better benchmarks/evaluations, and better training objectives. The call to research by the authors goes beyond just technical improvements to discuss the role that design and governance plays as well. This is a well thought out set of recommendations that can help future researchers find interesting directions.
3. **Interesting Description of the Roles AI Plays** - One of the authors key ideas is that adjusting AI based on the role it plays is critical in optimizing for well-being. The authors propose different roles such as a "concierge," "collaborator," and a "coach." Each of these provides different types of information based on the situation, and avoids inappropriate responses due to the user looking for one thing and the AI model providing something completely different.

## Weaknesses
1. **Unclear why exactly preference learning is the issue** - The first argument made by the author is that preference learning is inappropriate to optimize for well-being. While the argument is understandable, it does not clearly resolve the issue. That is, it is not clear whether any type of preference learning/any preference learning algorithm will suffer from issues such as sycophancy, or whether it is just our current paradigm of short-horizon preferences that suffers from this. This distinction is important, as it is unclear whether the authors are proposing that we investigate a fully new paradigm for well-being, or simply modify the existing learning paradigm.

**Support:**

3

---

> ### Author Rebuttal · Authors · 2026-03-31
>
> We thank the reviewer for the strong endorsement and for highlighting the paper’s clear argument structure, concrete next steps, and role-based interaction design. We especially appreciate the distinction you raise below, as it helped us sharpen the paper’s central claim.
>
> **Whether the issue is preference learning itself or the short horizon (W1).** This is exactly the right distinction, and we sharpen it in revision. Our position is about the horizon, not the paradigm. Preference learning with longer-horizon signals is exactly what we're calling for. The problem isn't RLHF per se, it's RLHF on single-turn preference labels without long-horizon outcomes in the loop. We've tightened the language throughout to make it clear that the call is additive (supplement preference optimization with longer-horizon signals; don't replace it).
>
> **How do we know sycophancy is a symptom of short-run preferences? (Q1).** We do not claim that well-being optimization would eliminate sycophancy entirely. The narrower claim is that it would reduce one important incentive for a specific form of sycophancy in socioemotional settings: affirming questionable framings, because affirmation scores well on immediate feedback and engagement-like signals. Other sources of sycophancy (e.g., training-data distributions, instruction-following biases, or deployment cues) would still require complementary interventions. But current evidence also points toward the training signal mattering. E.g., non-RLHF base models show less sycophancy than their post-trained counterparts (Sharma et al. 2023), and Cheng et al. (2025) show that sycophantic responses are systematically rewarded in preference datasets.
>
> In revision, we also add a worked example (a user repeatedly seeking reassurance about ambiguous romantic signals) to make this concrete — the preference-optimized response ("They're probably just busy!") scores well on single-turn evaluation but causes trajectory-level harm over weeks. The example traces how three technical approaches (trajectory-level comparison, potential-based reward shaping, theory-grounded per-turn proxies) would generate different training signals that reward honesty over reassurance.
>
> [1] Sharma et al., Towards Understanding Sycophancy in Language Models, 2023.
>
> [2] Cheng et al., Sycophantic AI Decreases Prosocial Intentions and Promotes Dependence, Science / arXiv:2510.01395.

---

> > ### Author Rebuttal · Reviewer_4eZ4 · 2026-04-01
> >
> > I think this is a very interesting position paper that will generate lots of good discussion; I hope it will be accepted.

---

### Official Review · Reviewer_uHfv · 2026-03-09

**Significance:** 3
**Argument Clarity:** 2
**Rating:** 5
**Confidence:** 2

**Questions:**

I have some second-thoughts as to how much the problem brought forward is actually a "technical ML" problem versus it being more a governance/business/political problem since the incentives of the companies producing LLMs (ie., maximise usage of the LLMs they serve since this maximises their profit) are not aligned with the incentives of the individuals (ie., use LLMs in such a way that benefits me the most which often might be to not use an LLM). So I wonder if it's really the technology we are lacking here or more the motivation to deploy the existing tools in such a way that puts user's needs first instead (analogy that comes to mind here is social media and also how it can be seen as another example of a technology that might not be "optimised for our well-being")

**Alternative Views Section:**

Yes

**Compliance With Llm Reviewing Policy A Conservative:**

Affirmed.

**Discussion Potential:**

3

**Final Justification:**

The authors addressed my main outstanding concerns during the rebuttal, hence I maintain my overall positive recommendation.

**Paper Summary:**

The submitted manuscript talks about the problem of how LLMs that have been optimised with short-term rewards in mind (via RLHF) are increasingly being used for long-term tasks (like giving out general life advice) where short and long term incentives could diverge. The authors make an argument that "shallow" fixes like system prompts with guardrails are not enough to mitigate this issue and make a call to action to rethink also the (post)-training and benchmarking/evaluation of general-purpose LLMs/agents.

I like the paper overall as it tries to tackle a very important societal question that is very relevant due to ever-increasing reliance on LLMs. Though I do think the paper could be a bit more concrete in how the ML community should go about resolving those problems

I would add that I am not an alignment researcher, so I'm also not that familiar with the related literature in this space. Hence I'm giving my review a low confidence (2)

**Position:**

Yes

**Position In Title:**

Yes

**Related Work:**

2

**Strengths And Weaknesses:**

Strengths:
- The paper discusses a very important and consequential problem in deploying general-purpose LLMs and it will hopefully lead to more discussion/progress in this area
- Even though the paper deals with some hard-to-define concepts like "well-being", I feel like the authors do a good job of not being too subjective and remain sound/objective
- Unlike some other papers I reviewed in the position track, this paper I think actually fits the position track quite well. Their position is clear and they argue for it an easy-to-follow way

Weaknesses:
- I think Call to Action could be made more concrete. For example in section 6.3 the authors argue for things like multi-objective RL, longer-term horizons, trajectory-level rewards etc. All those areas have been widely studies in the RL literature, so the authors could mention the exact methods they have in mind (together with citing the most relevant papers). This way they would go from high-level/abstract call to action, to a more actionable/practical one
- One of the solutions proposed is to have more explicit "personas" e.g. coach vs collaborator vs concierge. But I wonder if this is enough because maybe sometimes users might anyways default to using a single persona because they find it more agreeable/soothing etc. So to really prevent the negative delusional spirals mentioned in the paper I guess we should also think about having some real-time monitoring (most likely via another LLM/agent) in place that would try to detect the conversation drifting into an unhealthy/toxic direction

**Support:**

2

---

> ### Author Rebuttal · Authors · 2026-03-31
>
> We thank the reviewer for the supportive review and for recognizing that the paper tackles an important societal question, remains objective despite dealing with hard-to-define concepts, and fits the position track well. We also appreciate the concrete suggestions for making the paper more actionable. We address each point below.
>
>
> **More concrete call to action (W1).** We agree, and in the revision, we ground the technical directions in a worked example. Consider a user who keeps coming back for reassurance about ambiguous romantic signals — they've been on a few dates, the other person has been responding intermittently, and the user wants validation. A preference-optimized model reliably says, "They're probably just busy — don't overthink it!" Repeated over weeks, the trajectory-level harm (delayed self-assessment, months of false hope) never shows up in a single-turn evaluation.
>
> We now describe three approaches at increasing data cost:
> (1) trajectory-level comparison, rewarding turns in conversations that ended in positive user-reported outcomes over negative ones;
> (2) potential-based reward shaping (Ng et al. 1999), training a value model to predict long-term outcomes from partial traces and using it to shape per-turn rewards;
> (3) theory-grounded per-turn proxies — operationalizing behavioral markers psychology links to well-being (e.g., helping users sit with ambiguity vs. resolving it prematurely). These are also grounded in the specific RL literature the reviewer rightly notes exists.
>
>
> **Real-time monitoring (W2).** Good idea. This aligns well with our graduated interventions proposal (nudge → challenge → escalation/handoff). An LLM monitor that detects conversational drift toward unhealthy dynamics is a natural implementation of the escalation layer. We note this as an additional direction in revision. It also connects to the "Who" tension; a monitoring agent could enforce collective constraints (e.g., flagging patterns associated with delusional reinforcement across users) that a per-user optimization loop would miss.
>
>
> **Technical vs. governance problem (Q1).** We think it is genuinely both, and your social-media analogy is exactly the right one. The technical side matters because the field still lacks the right objectives, evaluation protocols, and affordances for long-horizon socioemotional use; the governance/business side matters because short-term engagement and approval incentives can overwhelm those tools in deployment. Product-layer or policy fixes will be unstable if the underlying model continues to be optimized mainly for short-horizon approval, while governance alone is weak unless there are technical affordances that make long-horizon outcomes measurable and auditable. The technical and governance agendas are complementary, not competing. The ML community is best positioned to move on the technical side, which is why we frame the paper that way for ICML.

---

> > ### Author Rebuttal · Reviewer_uHfv · 2026-04-02
> >
> > I thank the authors for their response. I still think the paper could be more precise/concrete when talking about technical proposals (as raised by some other reviewers too), but I also believe that the paper can serve as a good starting point for interesting discussions on a very important topic for ML community, hence I maintain my positive overall assessment.

---

### Official Review · Reviewer_78Hu · 2026-03-10

**Significance:** 4
**Argument Clarity:** 3
**Rating:** 3
**Confidence:** 4

**Questions:**

1. What, concretely, is the paper’s causal claim? Is it that current post-training objectives are the primary driver of the cited harms, or that they are one important contributor among several interacting factors? Can the authors more clearly separate what is supported by direct empirical evidence from what is speculative extrapolation or conceptual argument?

2. How do the authors define “well-being” at a level precise enough to support system design and evaluation, rather than only philosophical discussion? Among the proposed targets (e.g., reduced regret, sustained goal progress, improved self-regulation, better relationship outcomes, improved belief accuracy), which are primary, which are proxy measures, and how should they be prioritized when they conflict?

3. What is the paper’s genuinely new contribution beyond synthesizing existing concerns from AI alignment, AI safety, recommender systems, mental-health chatbot risk, and value pluralism? Could the authors make the contribution more decisive by articulating a clearer formalism, sharper taxonomy, or more concrete agenda that goes beyond a list of sensible recommendations?

4. Can the authors sharpen the paper’s position into a more precise question such as: which preferences, from whom, over what horizon, and under what constraints? More specifically, (1) which preferences do the authors regard as relevant: immediate expressed preferences, reflective preferences, longitudinal preferences, community-informed preferences, or some combination? (2) From whom should these preferences be elicited: the end user, domain experts, affected third parties, or broader social stakeholders? (3) Over what horizon should preference satisfaction be evaluated, and under what constraints should it be overridden or regularized?

5. Can the authors give a more concrete account of what trajectory-level rewards would look like in this setting, including what signals would be used and how delayed outcomes would be attributed to model behavior? What specific constrained optimization setup do the authors have in mind, and what would the constraints be?

6. How do the authors propose validating this "well-being mode" without introducing severe privacy concerns, confounding from user self-selection, or low-quality feedback loops?

**Alternative Views Section:**

Yes

**Compliance With Llm Reviewing Policy A Conservative:**

Affirmed.

**Discussion Potential:**

3

**Final Justification:**

The rebuttal has resolved part of my concerns, please refer to the Rebuttal Acknowledgement for details.

**Paper Summary:**

The paper's claim is that when LLMs are used for socioemotional support and life guidance, optimizing them primarily for short-horizon preference signals is the wrong objective. The authors argue that this induces predictable failure modes, including unjustified affirmation, over-accommodation, and trajectory-level harms that standard single-turn evaluations miss. The paper’s main contribution is therefore conceptual: it reframes the problem around three tensions: immediate vs. long-horizon outcomes, individual vs. collective benefit, and user choice vs. AI guidance. It calls for new training objectives, longitudinal evaluation, product affordances, and governance mechanisms tailored to “well-being modes”.

**Position:**

Yes

**Position In Title:**

Yes

**Related Work:**

3

**Strengths And Weaknesses:**

**Strengths**

1. The paper addresses an important and timely question. The shift from LLMs as productivity tools to companions, advisors, and socioemotional interlocutors is real, and the manuscript is correct that alignment assumptions that may be tolerable in coding or summarization can become much more problematic in life-guidance settings. That basic framing is valuable and worthy of discussion.

2. The “When / Who / How” framework is clear, memorable, and useful as a discussion scaffold. Table 1 gives the paper a strong backbone by linking present incentives, likely failure modes, proposed interventions, and candidate tests. For a position paper, this is one of the better aspects of the submission.

3. The paper also does a reasonable job of anticipating objections. Section 7 on alternative views acknowledges concerns about paternalism, privacy, and whether product-layer interventions might suffice, which makes the piece more balanced than a purely polemical essay.

**Weaknesses**

1. My main concern is that the paper’s evidentiary basis is too weak for the strength and breadth of its claims. The central thesis is plausible, but much of the argument rests on a mixture of anecdotal reporting, broad intuitions about incentives, and selective references to emerging studies. The paper does not concretely establish that current LLM post-training objectives are in fact the primary driver of the cited harms, as opposed to a more complicated interaction among deployment choices, prompting, safety policy, product UX, user selection effects, and domain-specific misuse. The paper is strongest as a hypothesis-generating essay, but it is written as though the causal story is already established.

2. The paper overstates how settled the construct of “well-being alignment” is. It acknowledges that well-being is philosophically contested, but then proceeds as if this acknowledgement is enough to operationalize the concept. It is not. The proposed downstream targets stated in Lines 127-129: reduced regret, sustained goal progress, improved self-regulation, better relationship outcomes, improved belief accuracy, are heterogeneous, difficult to aggregate, and frequently in tension with one another. The manuscript does not provide a convincing account of how these should be measured, traded off, personalized, or audited across cultures and use cases. As written, “optimize for well-being” risks becoming a rhetorically attractive but technically underdetermined slogan.

3. The paper under-specifies what is genuinely new relative to existing discussions of AI alignment, AI safety, recommender-system externalities, mental-health chatbot risk, and value pluralism. The manuscript assembles these threads into a coherent narrative, but the novelty is mostly synthetic. This is acceptable only if the synthesis is exceptionally sharp or generative. Here, I do not think it yet reaches that bar. The proposed directions including: longitudinal evaluation, multi-objective optimization, role selection, transparency, audits, governance, are sensible, but also fairly expected once one accepts the framing. The paper does not crystallize a particularly surprising thesis, formal framework, or research agenda that would decisively move the discussion forward.

4. The paper's argument somewhat one-sided in how it treats preference optimization. It frames RLHF-like training largely as an engine of sycophancy and short-term approval seeking, but it gives insufficient attention to the fact that user preference modeling may be a necessary ingredient for many forms of respectful assistance, including forms of well-being support that must remain non-coercive. Put differently, the paper argues against optimizing solely for immediate approval, but too often rhetorically collapses that into a broader indictment of preference optimization itself. **The more interesting question is not whether preferences are flawed—they obviously are—but which preferences, from whom, over what horizon, and under what constraints. That sharper formulation is missing.**

5. There exists a mismatch between the paper’s ambition and its level of technical precision. The manuscript repeatedly invokes concepts such as trajectory-level rewards, constrained optimization, subgroup audits, and update-drift tracking, but these are gestured at rather than developed. For example, the call for an opt-in “well-being mode” sounds reasonable at the product level, yet the paper does not explain how such a mode would avoid simply becoming another vague persona layer on top of an unchanged base model, nor how its success would be validated without introducing major privacy, confounding, and feedback-quality problems. In a strong position paper, even if the full solution is not known, the proposed route should be sharper than a list of good intentions.

**Support:**

2

---

> ### Author Rebuttal · Authors · 2026-03-31
>
> We appreciate the careful engagement. We will revise the paper to answer the core concerns more precisely.
>
> **Causal claim and evidentiary standard (Q1, W1).** We sharpen the causal claim that post-training objectives are one important contributor among several interacting factors, not the sole driver. However, it is the factor most directly under ML researchers' control, which is why it merits focused attention at ICML.
>
> The reviewer notes the paper is "strongest as a hypothesis-generating essay." We think this *is* the function of the position track. The CFP asks for well-argued positions that stimulate productive discussion, not settled causal proof. We cite converging evidence to motivate an actionable hypothesis.
>
> **Novelty (Q3).** An LLM could be AI-safe, aligned, mental-health-conscious, and value-pluralism-aware, and still optimize for short-term engagement. No existing thread isolates long-horizon well-being as a specific gap in post-training objectives. Our When/Who/How framework makes this problem tractable by decomposing it into independently addressable tensions.
>
> Notably, the reviewer's own sharpened formulation "which preferences, from whom, over what horizon, under what constraints" maps directly onto our framework: *When* = over what horizon, *Who* = from whom, *How* = under what constraints. We take this convergence as evidence that the taxonomy is doing useful work.
>
> **Preference optimization (Q4, W4).** We agree the rhetoric sometimes collapses "short-horizon preference optimization" into "preference optimization" generally, and correct this. Our call is additive. Preference optimization is necessary and effective for many uses but insufficient in well-being contexts. Preference learning with long-horizon outcomes in the loop is exactly what we advocate.
>
> **Technical precision (Q5, W5).** In revision, we ground the technical directions with a worked example. Consider a user who keeps coming back for reassurance about ambiguous romantic signals — they've been on a few dates, the other person has been responding intermittently, and the user wants validation that things are going well.
>
> *Current failure mode.* The preference-optimized model generates the high-approval response: "They're probably just busy… don't overthink it!" Repeated over weeks, this delays honest self-assessment, extends emotional investment in a likely dead end, and erodes the user's trust in their own judgment. The trajectory-level harm is invisible to single-turn evaluation.
>
> We now describe three approaches and show how they apply to this example:
>
> (1) *Trajectory-level comparison.* Compare conversations ending in positive outcomes ("I'm glad I moved on early") vs. negative ones ("I wish I'd seen the signs"). Reward turns in good-outcome conversations more. Optionally use LLM reasoning to assign differential credit to specific turns. Skips fine-grained credit assignment but shifts the distribution.
>
> (2) *Potential-based reward shaping.* Train a value model to predict long-term outcomes from partial traces, then shape per-turn rewards using value estimates. Following Ng et al. (1999) with theoretical guarantees. Requires more data; frontier labs plausibly have it, others could use synthetic trajectories.
>
> (3) *Theory-grounded per-turn proxies.* Operationalize behavioral markers that psychological theory links to well-being, such as whether the assistant helps the user sit with ambiguity vs. resolving it prematurely, and encourages perspective-taking vs. confirmation. Cheap to evaluate per-turn; requires interdisciplinary work to validate.
>
> These form a cost-signal gradient: cheap proxies (#3), medium-cost value models (#2), expensive longitudinal outcomes (#1). This is illustrative. Working out exact implementations is part of what we call for.
>
> **Operationalizing well-being (Q2, W2).** We do not claim to resolve the operationalization problem. We claim it must be stated and approximated rather than ignored. The worked example shows how imperfect proxies (user-reported regret, goal progress) ground concrete design. Validated instruments (PERMA profiler, Satisfaction with Life Scale) serve as lightweight follow-up measures, and the 3-tension framework scaffolds deciding which proxies matter in which contexts.
>
> **Privacy and validation (Q6).** Frontier labs already collect extensive interaction data to optimize engagement. Opt-in follow-ups for well-being signals are strictly less invasive than current practice. The incremental privacy concern is valid. Self-selection confounding is addressable via longitudinal A/B experimentation, the same methodology used for product decisions. The question is not feasibility, but whether we direct existing measurement toward well-being rather than only engagement.

---

> > ### Author Rebuttal · Reviewer_78Hu · 2026-04-04
> >
> > I have read the authors’ rebuttal carefully. The rebuttal resolves part of my concerns, thus I would like to increase my rating to 3.
> >
> > Specifically, the rebuttal improves the paper’s framing by clarifying that post-training objectives are one contributor among several, and by distinguishing short-horizon preference optimization from preference optimization more broadly. It also provides a more concrete worked example. However, these additions do not materially change my overall assessment. The paper still relies primarily on a plausible conceptual narrative rather than strong support for its central claims, and the main open issues I raised, especially around operationalizing well-being, validating the proposed framework, and clarifying what is genuinely new beyond synthesis, remain only partially addressed. I therefore still regard the paper as interesting and timely, but not yet strong enough for clear acceptance in this track.

---

### Official Review · Reviewer_sJrL · 2026-03-13

**Significance:** 3
**Argument Clarity:** 2
**Rating:** 4
**Confidence:** 3

**Questions:**

Questions:

1. The paper mentions that RLHF models are "approximately 40% more likely to reinforce incorrect beliefs" than non-RLHF versions. Are there any details on the specific benchmark or dataset used to reach this figure?

**Alternative Views Section:**

Yes

**Compliance With Llm Reviewing Policy A Conservative:**

Affirmed.

**Discussion Potential:**

3

**Final Justification:**

The authors have fully addressed my concerns. I look forward to seeing all of the promised additions.

**Paper Summary:**

The authors argue that RLHF makes models fluent and helpful for productivity, but it also induces a sycophantic attractor where models avoid necessary friction to keep users satisfied in the moment. The paper advocates for a shift where LLMs should be optimized for human flourishing, especially those used for life guidance. This does not mean replacing the default mode for productivity tasks but creating an explicit, opt-in "well-being" mode governed by long-horizon outcomes.

**Position:**

Yes

**Position In Title:**

Yes

**Related Work:**

2

**Strengths And Weaknesses:**

Strengths

1.	The paper provides a highly organized roadmap centered around three primary tensions: time horizon, beneficiary, and relational stance. This structure allows for a principled analysis of why current optimization fails in socioemotional contexts.
2.	The paper also provides an actionable framework in the sense that, rather than remaining purely philosophical, it proposes concrete directions, including incorporating longitudinal metrics (like 1–4 week follow-ups), multi-objective training frameworks, and explicit product interaction modes.

Weaknesses

1.	The paper calls for trajectory-level rewards and multi-objective post-training, but it remains relatively high-level in terms of the specific mathematical or architectural implementations required to address the sparse credit-assignment problem.
2.	The term well-being is somewhat vague and difficult to measure. I think the challenge here is how to translate this broad philosophical concept into dense, legible training signals for gradient-based optimization. If the authors can address or motivate this, it would substantially increase the paper's impact.
3.	The authors advocate for longitudinal measurement but acknowledge the significant privacy risks. It is more beneficial to have a more detailed exploration of how privacy-preserving techniques could realistically support the high data density needed for RLHF, like federated learning or edge aggregation.

**Support:**

2

---

> ### Author Rebuttal · Authors · 2026-03-31
>
> We thank the reviewer for the careful review. We appreciate that the reviewer found the paper well organized, the three-tension roadmap helpful, and recognized that the paper aims to be actionable rather than purely philosophical.
>
> **Technical implementation (W1).** In revision, we ground the technical directions with a worked example. Consider a user who keeps coming back for reassurance about ambiguous romantic signals, they've been on a few dates, the other person responds intermittently, and the user wants validation that things are going well. A preference-optimized model reliably says, "They're probably just busy — don't overthink it!" Repeated over weeks, this delays honest self-assessment, extends emotional investment in a likely dead end, and erodes the user's trust in their own judgment. Single-turn evaluation never sees this.
>
> We now ground three technical approaches at increasing data cost:
>
> (1) *Trajectory-level comparison.* Compare conversations ending in positive user-reported outcomes ("I'm glad I moved on early") vs. negative ones ("I wish I'd seen the signs"). Reward turns in good-outcome conversations more. Optionally use LLM reasoning to assign differential credit to specific turns. This skips fine-grained credit assignment but shifts the distribution.
>
> (2) *Potential-based reward shaping.* Train a value model to predict long-term outcomes from partial conversation traces, then shape per-turn rewards using value estimates. This follows Ng et al. (1999), which provides theoretical guarantees that the optimal policy is preserved. Requires more data. Frontier labs plausibly have it; others could use synthetic trajectories or enrichment.
>
> (3) *Theory-grounded per-turn proxies.* Operationalize behavioral markers that psychology links to well-being. For e.g., whether the assistant helps the user sit with ambiguity vs. resolving it prematurely, encourages perspective-taking vs. confirmation. Cheap to compute per-turn but requires interdisciplinary work to validate.
>
> We intentionally present a gradient because the right approach depends on data availability, which varies enormously across settings. The sparse credit-assignment problem here is real and hard; we're writing this position paper to argue that it's important to solve, not to claim we've solved it.
>
> **Well-being measurement (W2).** We agree that well-being needs to be more concrete as a training signal. In the example above, the relevant signals include whether the user reported regret weeks later. Did they make progress on their stated goals? Did they feel the assistant helped them think clearly, or just feel good? Validated instruments, such as the PERMA Profiler and the Satisfaction with Life Scale, already exist for this purpose. The claim is that such measures should be integrated into the training and evaluation loop, not merely used in post hoc studies.
>
> **Privacy (W3).** We agree privacy deserves more detail. Our claim is not that longitudinal measurement is free of privacy cost, but that it can be designed to be less intrusive than logging dense raw transcript data. In revision, we will discuss opt-in sparse follow-ups, on-device or aggregated features, and federated-update style approaches as concrete privacy-preserving directions.
>
> **40% statistic (Q1).** We agree this needs to be more precise. In revision, we will replace the abstract’s imprecise “approximately 40%” phrasing with the specific underlying results. The clearest support here comes from Ibrahim et al. (2025), who report that warm/empathetic tuning makes models about 40% more likely than their original counterparts to reinforce incorrect user beliefs, and that they exhibit higher overall error rates on safety-critical tasks.
>
> **Argument clarity and related work.** We have tightened the argument in several ways: (a) sharper causal claim: post-training is one contributor among several, not the sole driver, but the one most under ML researchers' control; (b) additive framing: we're not arguing against preference optimization, we're arguing it's insufficient in well-being contexts; (c) additional related work grounding the three technical approaches in established RL literature (Ng et al. 1999 for reward shaping, standard trajectory-level RL for credit assignment).
>
> Ibrahim et al., Training language models to be warm and empathetic makes them less reliable and more sycophantic, 2025.

---

> > ### Author Rebuttal · Reviewer_sJrL · 2026-04-04
> >
> > The authors have fully addressed my concerns. I raise my score to 4.

---

### Decision · Program_Chairs · 2026-04-30

**Decision:**

Accept (regular)

**Comment:**

The authors take the position that we need large language models optimized for our well-being

The reviewers agreed that the paper has several strong points:

* The paper addresses an important and timely question. The shift from LLMs as productivity tools to companions, advisors, and socioemotional interlocutors is real. Alignment assumptions that may be tolerable in coding or text summarization can become more problematic in life-guidance settings.

* The reviewers agreed that the main question is valuable and worthy of discussion.

* One reviewer noted that Table 1 provides a strong backbone for the paper by linking present incentives, likely failure modes, proposed interventions, and candidate tests.

* The paper discusses alternative views and possible objections.

* The paper provides an actionable framework. It outlines clear next steps and a call for further research.

Most of the reviewers were positive about this paper, but one of the reviewers noted that
"I therefore still regard the paper as interesting and timely, but not yet strong enough for clear acceptance in this track."

One of the reviewers gave the paper a “strong accept” rating.

The AC has read the authors’ rebuttals and comments and has incorporated them into the decision-making process.